# Progression of type 1 diabetes from latency to symptomatic disease is predicted by distinct autoimmune trajectories

Bum Chul Kwon [1,10 ✉], Vibha Anand [1,10], Peter Achenbach [2], Jessica L. Dunne [3], William Hagopian [4], Jianying Hu [5], Eileen Koski [5], Åke Lernmark [6], Markus Lundgren [6], Kenney Ng [1], Jorma Toppari [7], Riitta Veijola [8], Brigitte I. Frohnert[9] & the T1DI Study Group*

Development of islet autoimmunity precedes the onset of type 1 diabetes in children, however, the presence of autoantibodies does not necessarily lead to manifest disease and the onset of clinical symptoms is hard to predict. Here we show, by longitudinal sampling of islet autoantibodies (IAb) to insulin, glutamic acid decarboxylase and islet antigen-2 that disease progression follows distinct trajectories. Of the combined Type 1 Data Intelligence cohort of 24662 participants, 2172 individuals fulfill the criteria of two or more follow-up visits and IAb positivity at least once, with 652 progressing to type 1 diabetes during the 15 years course of the study. Our Continuous-Time Hidden Markov Models, that are developed to discover and visualize latent states based on the collected data and clinical characteristics of the patients, show that the health state of participants progresses from 11 distinct latent states as per three trajectories (TR1, TR2 and TR3), with associated 5-year cumulative diabetes-free survival of 40% (95% confidence interval [CI], 35% to 47%), 62% (95% CI, 57% to 67%), and 88% (95% CI, 85% to 91%), respectively (p < 0.0001). Age, sex, and HLA-DR status further refine the progression rates within trajectories, enabling clinically useful prediction of disease onset.

[1] Center for Computational Health, IBM Research, Cambridge, MA, USA. [2] Institute of Diabetes Research, Helmholtz Zentrum München, German Research Center for Environmental Health, Munich-Neuherberg, Germany. [3] JDRF, New York, NY, USA. [4] Pacific Northwest Research Institute, Seattle, WA, USA. [5] Center for Computational Health, IBM Research, Yorktown Heights, NY, USA. [6] Department of Clinical Sciences Malmö, Lund University CRC, Skåne University Hospital, Malmö, Sweden. [7] Institute of Biomedicine and Centre for Population Health Research, University of Turku, and Department of Pediatrics, Turku University Hospital, Turku, Finland. [8] University of Oulu and Oulu University Hospital, Department of Pediatrics, PEDEGO Research Unit, Oulu, Finland. [9] University of Colorado, Denver, CO, USA. [10] These authors contributed equally: Bum Chul Kwon, Vibha Anand. * A list of authors and their affiliations appears at the end of the paper. ✉email: bumchul.kwon@us.ibm.com

Recent advances in type 1 diabetes research have increased appreciation of heterogeneous patterns of islet auto-immunity before a diagnosis of clinical diabetes. Previous research points to two distinct pathways from the appearance of islet autoimmunity to clinical diabetes, i.e. associated with initial development of islet autoantibodies (IAb) to either insulin (IAA) or glutamic acid decarboxylase (GADA)[1]. There is evidence that these pathways may be triggered by pathogenic exposures, with differential associations for intestinal viruses or prenatal exposures[2–4]. Further, there are observed differences in both genetic associations and expansion from initial IAb to multiple autoantibodies and risk of progression to diabetes based on the pattern of autoantibody acquisition[3,5–7].

While the number of observed IAb predicts risk for progression to type 1 diabetes, the temporal progression of these biomarkers displays heterogeneous patterns and could further stratify risk[7–11]. Previous observations underscore the need to better define individual trajectories from islet autoimmunity to type 1 diabetes[12]. Better identification and understanding of the heterogeneity of the disease may have substantial implications for elucidation of its etiology. Further, the ability to predict diabetes risk, progression rate, and intervention response may enable personalized therapeutic approaches.

To better understand these patterns, we investigated the presence or absence of three islet autoantibodies, GADA, IAA, and islet antigen-2 (IA-2A), prior to clinical diabetes in a large cohort of data combined from five prospective studies in four countries. Using an unsupervised machine learning approach, we generated quantitative descriptions of underlying progression patterns from islet autoimmunity to diagnosis of type 1 diabetes and utilized novel visualization strategies to gain new insights into differences between individuals in these trajectories.

## Results

**Three trajectories.** Continuous-Time Hidden Markov Models (CT-HMMs) were learned as disease progression models (DPM) based on presence or absence of IAb from the longitudinal T1DI study cohort. Using machine learning methods, a model containing 11 latent states was discovered that best fit the observed data and was subsequently applied to all IAb positive participants to draw the insights presented here. For the 643 diagnosed (D) participants, the discovered states formed three trajectories, TR1, TR2, and TR3, each characterized by a distinct sequence of latent states (Fig. 1a), which were further explored using interactive data visualization and statistical analysis.

In this figure, each latent state is described by a set of probabilities for presence of each IAb (Fig. 1a). Longitudinal observation sequences of participants with IAb positivity were labeled using the model, and these participants were then divided into those who developed type 1 diabetes (Diagnosed/D) during the study period vs. those who did not or were lost to follow-up (Undiagnosed/UD). Statistical analyses correlated these latent states and resulting trajectories with other study variables to draw insights from all participants with IAb positivity about each of these two groups.

**Diagnosed participants.** As shown in Fig. 1a, each trajectory starts with an initial state with low IAb probability (TR1-0, TR2-0, TR3-0), indicating most diagnosed individuals begin without any positive IAb. After the initial state with low probability, a unique sequence of IAb progression follows in each trajectory. Participants in TR1-0 progress to TR1-1, which has a high probability of multiple IAb positivity: GADA (0.93), IAA (0.62), and IA-2A (0.94) followed by TR1-2, which has 100% IA-2A positivity and loss of GADA (0.16) and IAA (0.04).

In trajectory TR2, participants added a single IAb at a time, most typically in the following order: IAA (0.86) in TR2-1, and IA-2A (0.98) in TR2-2. While almost all had acquired GADA (0.99) in TR2-3, there was a subset of individuals who were already positive for GADA in TR2-1 (0.58) or TR2-2 (0.26). Some individuals then progressed to the multiple positive antibody state TR2-3 (GADA 0.99, IAA 0.96, and IA-2A 1.0) and then to TR2-4, where they lost IAA (0.08) while maintaining a high probability of GADA (0.97) and certainty of IA-2A (1.00).

Similarly, most participants in trajectory TR3 added IAb in series as follows: GADA (0.98) in TR3-1, and IA-2A (1.00) in TR3-2 (multiple IAb positive). A minority were positive for IAA in TR3-1 (0.21), but most were no longer IAA positive in TR3-3 (0.07).

For those diagnosed (Fig. 2a), TR3 shows a later onset of islet autoimmunity than the other trajectories, with a median TR3-1 entry age of 3.3 years, compared to TR1-1 (2.5 years) and TR2-1 (1.3 years). The diagnosed participants in TR2 stay in the first three states (TR2-0, TR2-1, TR2-2) briefly, as illustrated by the widths of the respective state nodes in Fig. 2a. The diagnosed participants in TR2 were diagnosed at states ranging from TR2-1 to TR2-4, among which the numbers were fairly evenly distributed, while those in TR1 had a higher diagnosis rate in state TR1-1 and those in TR3 were disproportionately diagnosed in the final state, TR3-2, i.e., after gaining IA-2A as an additional autoantibody.

**Undiagnosed participants.** The 1502 undiagnosed (UD) participants (Fig. 1b) followed three trajectories - TR1(483), TR2 (257), and TR3 (762). Of the undiagnosed, 628 (42%) transitioned to IAb positive states from initial states; the rest stayed in IAb negative states, i.e., TR1-0, TR2-0, and TR3-0. Median entry age of the undiagnosed into IAb positive states is higher than that of the diagnosed participants for all states (Fig. 2b). For more details about trajectories, see Supplementary Discussion in Supplementary Information.

**Islet autoantibody pattern by age and trajectory.** As specific IAb patterns could exist in more than one trajectory, we examined the composition of trajectories amongst the seven possible IAb patterns across ages 2–7 years (Fig. 3a–f). We showed that for all but one pattern, the majority of individuals with that pattern were in a single dominant trajectory. For the single antibody positive patterns, proportions higher than 60% across these ages consisted of: TR3 for GADA + only (Fig. 3a), TR2 for IAA+ only (Fig. 3b), and TR1 for IA-2+ only (Fig. 3c). Among patterns with two IAb positive, both the GADA+ /IAA+ and the IAA+ /IA-2A+ patterns showed TR2 as a dominant trajectory (Fig. 3d, f, respectively).

Interestingly, GADA+ /IA-2A+ patterns showed that the dominant trajectory changes depending on participants' age: TR1 for ages 2–3, TR1 and TR3 for ages 3–4, and TR3 for ages 4–7 years (Fig. 3e). Participants who had three positive IAb belonged to TR2 more than other trajectories for all ages ≥ 3 years (Fig. 3g).

**Mean age at confirmed seroconversion and clinical onset.** The three trajectories showed significant differences in mean age at seroconversion and diagnosis (Table 1 and Supplementary Fig. 5). Among the diagnosed ($n = 546$), those in TR3 seroconverted significantly later ($F(2, 543) = 27.19$, $p < 0.0001$) than those in TR1 ($p = 0.001$) who seroconverted later than those in TR2 ($p = 0.001$). Among the undiagnosed ($n = 840$), those in TR3 seroconverted significantly later ($p < 0.0001$) than in TR1;

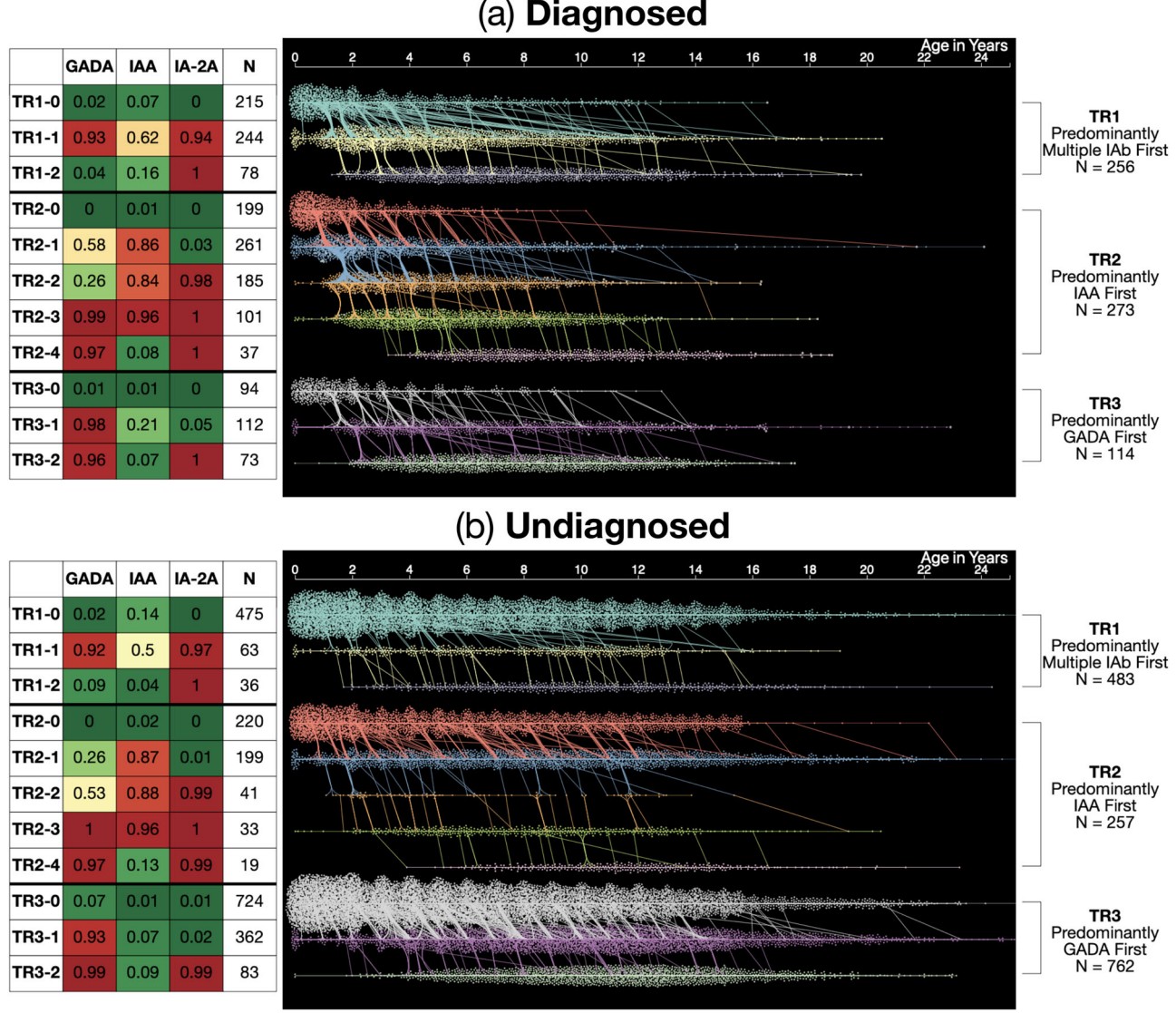

**Fig. 1 Three data-driven islet autoantibody trajectories towards type 1 diabetes.** The three trajectories discovered by the 11-state Hidden Markov model for (**a**) Diagnosed (D, 643 participants who were diagnosed with type 1 diabetes) and (**b**) Undiagnosed (UD, 1502 participants who were not diagnosed with type 1 diabetes during follow-up). Probabilities of islet autoantibody positivity for each of the 11 states are described in the table on the left, showing probabilities of each islet autoantibody for each state in text and color (heat map scale: green: 0, red: 1). Waterfall diagrams on right show visits (dots), HMM states (y-axis & color) of participants over time in years (x-axis). The three trajectories were characterized with respect to their first islet autoantibody positive states, respectively: TR1 Predominantly Multiple IAb first (256 P, 483 NP); TR2 Predominantly IAA first (273 P, 257 NP); TR3 Predominantly GADA first (114 P, 762 NP). Source data are provided as a Source Data file.

however, no significant differences were seen for seroconversion age between TR1 and TR2 or TR2 and TR3.

Among the diagnosed, participants in TR3 were significantly older ($F(2, 640) = 24.99$, $p < 0.0001$) than those in TR1 and TR2 at the time of diagnosis. Among the diagnosed, the age gap between confirmed seroconversion and clinical onset was longest for TR3 ($F(2, 543) = 8.93$, $p = 0.0002$) compared to TR1 ($p = 0.001$) and TR2 ($p = 0.037$), while there was no significant difference between TR1 and TR2 (Table 1).

**Sex and HLA-DR category.** Trajectory distributions by sex (Table 1) marginally differed across trajectories between diagnosed and undiagnosed ($X^2(5, n = 2145) = 10.59$, $p = 0.0602$). The pairwise comparison shows that diagnosed participants in TR3 had a higher ratio of females to males in comparison to the diagnosed participants in TR1 ($p = 0.0105$) and the diagnosed

participants in TR2 ($p = 0.0161$). No other pairs of trajectory/ diagnosis groups showed statistically significant differences in the ratio of female to male participants.

Finally, HLA-DR risk groups differed across trajectories between diagnosed and undiagnosed ($X^2(15, n = 2145) = 161.53$, $p < 0.0001$) (Table 1). The Chi-square test shows significant differences in the proportion of four HLA-DR risk groups among all groups of trajectories and diagnosis. Nine combinations of pairwise comparisons between the undiagnosed and the diagnosed in three trajectories all showed significant differences in the proportions of HLA-DR risk groups (all, $p < 0.0001$). The undiagnosed in TR1 were different from the undiagnosed in TR2 ($p = 0.0004$) and TR3 ($p = 0.0015$).

Complete results of the pairwise Chi-square tests for sex and HLA-DR are in Supplementary Tables 1 and 2, respectively, in the Supplementary Information.

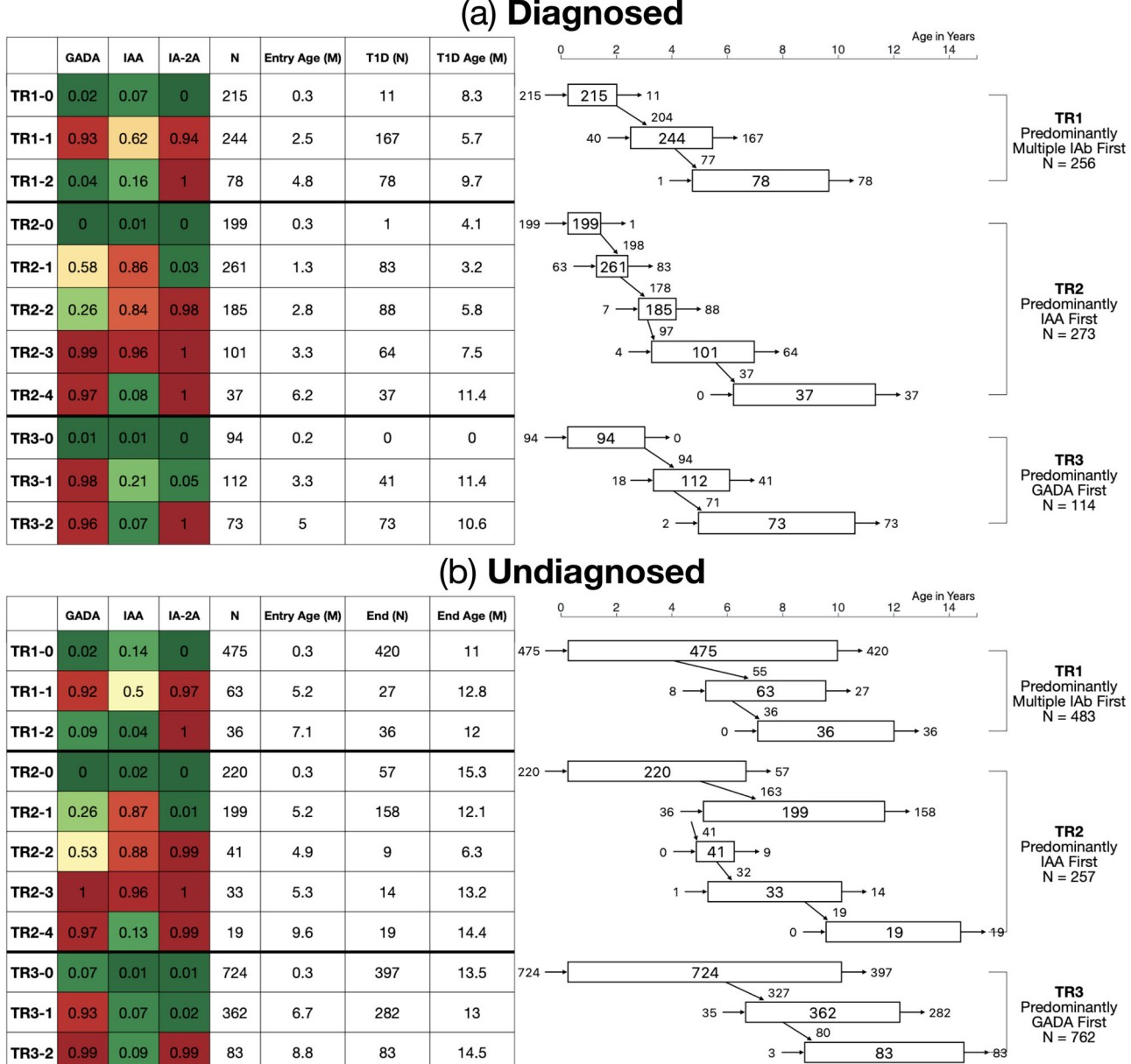

**Fig. 2 State transition diagram for diagnosed and undiagnosed participants in the three trajectories.** State transition diagram for D Diagnosed (**a**) and UD Undiagnosed (**b**). The table on left again shows probability of positivity for each islet autoantibody, total number of individuals in each state, median entry age (years) per state, number of individuals who are diagnosed (D) or lost in observation (UD) per each state and median age of type 1 diabetes diagnosis or last observation (years) for D and UD, respectively. Figure on right shows (i) how many participants enter each state (left-sided entry arrow); (ii) how many participants were diagnosed (D) or ended their observation (UD) at each state (right-sided exit arrow); (iii) how many participants progressed into subsequent states (diagonal transition arrow); (iv) how many participants are in each state (number in box). Each rectangle indicates median ages at the first (left side) and last (right side) observations at each state in years of age on the x-axis. Source data are provided as a Source Data file.

**Survival analysis.** Of the 2145 participants analyzed in both groups, 1241 (58%) had at least one follow-up visit after the initial IAb positive state (TR1-1, TR2-1, TR3-1). Survival analysis (Fig. 4) showed differences in progression to type 1 diabetes among trajectories after entry into initial IAb positive states. Participants in TR1 progress faster to diabetes than those in TR2, who progress faster than those in TR3 (Fig. 4, $p < 0.0001$). Post-hoc analysis showed significant differences in all pairwise comparisons between trajectories ($p < 0.0001$). The 5-year cumulative diabetes-free survival rates are significantly different (all pairwise differences, $p < 0.0001$): TR1 (mean = 40%; 95% Confidence Interval [CI]: 35 % to 47%), TR2 (mean = 62%; 95% CI: 57% to 67%), TR3 (mean = 88%; 95% CI: 85% to 91%). Of note, the mean ages of initial entrance to IAb positive states were significantly different overall ($p < 0.0001$); however, pairwise comparisons showed that while mean entry age in TR3 was later than TR1 ($p < 0.0001$) and TR2 ($p < 0.0001$), there was no difference in age of entry between TR1 and TR2 ($p = 0.0728$).

The survival curves stratified on sex are provided in Supplementary Fig. 3. Females in TR2 showed faster progression and lower rates of type 1 diabetes-free survival than males ($Z^2(1) = 5.7$, $p = 0.02$). There was no significant difference between sexes in TR1 ($Z^2(1) = 0.3$, $p = 0.6$) or TR3 ($Z^2(1) = 2.3$, $p = 0.1$).

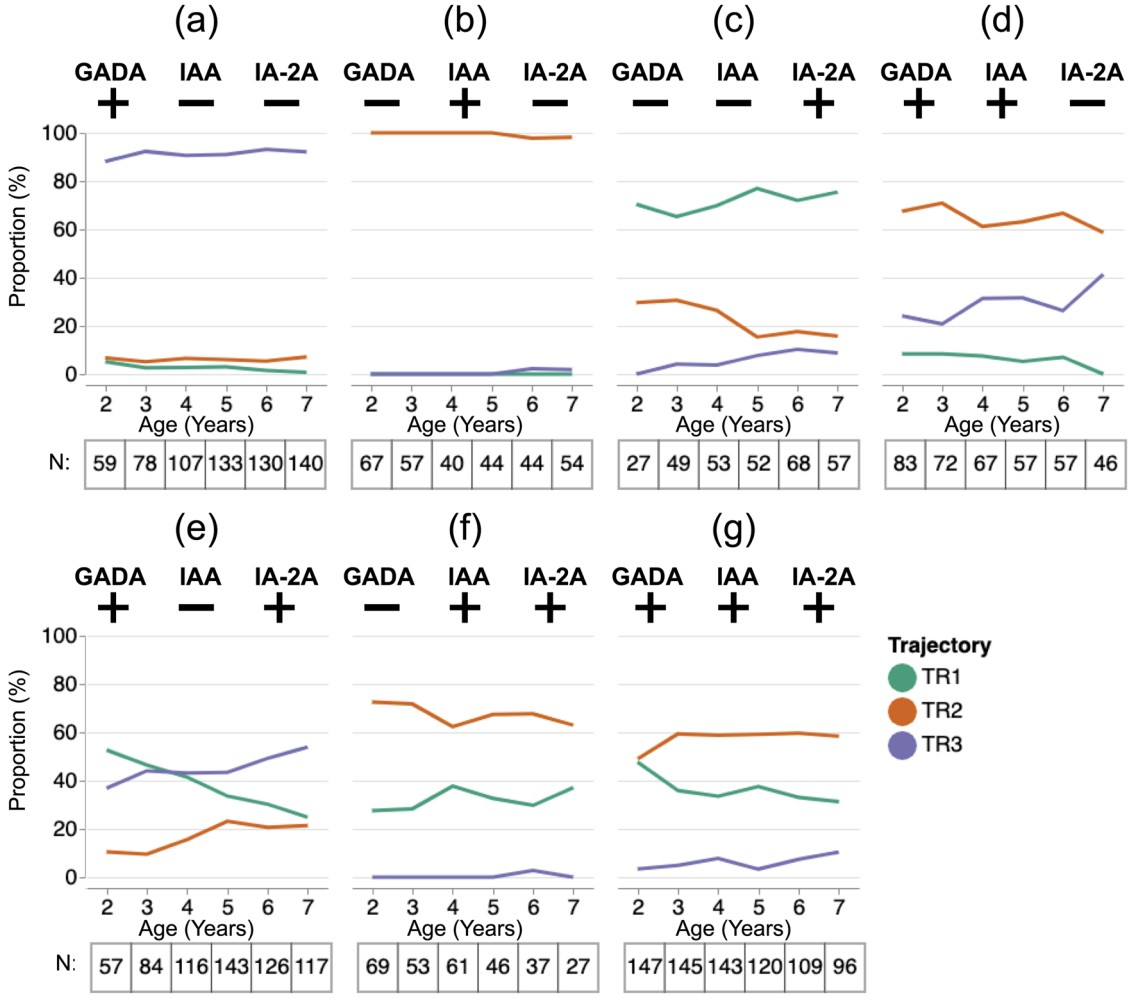

**Fig. 3 Proportion of participants in the three trajectories given islet autoantibody pattern and age.** Proportion (*y*-axis) of participants with each of seven possible IAb patterns (**a–g**) belonging to each trajectory (TR1: green, TR2: red, TR3: purple) over the ages between 2 and 7 (*x*-axis). Tables under each panel show the number of subjects who had the given IAb pattern at each corresponding age. Source data are provided as a Source Data file.

**Table 1 Distribution of undiagnosed and diagnosed participants in three trajectories over sex and HLA.**

| | | Undiagnosed | | | Diagnosed | | |
|---|---|---|---|---|---|---|---|
| | | TR1(*n* = 483) | TR2 (*n* = 257) | TR3 (*n* = 762) | TR1 (*n* = 256) | TR2 (*n* = 273) | TR3 (*n* = 114) |
| Sex | Male | 283 (59%) | 145 (56%) | 409 (54%) | 155 (61%) | 146 (53%) | 52 (46%) |
| | Female | 200 (41%) | 112 (44%) | 353 (46%) | 101 (39%) | 127 (47%) | 62 (54%) |
| | DR3/4 | 73 (15%) | 48 (19%) | 141 (19%) | 93 (36%) | 106 (39%) | 41 (36%) |
| HLA-DR status | DR4/X | 298 (62%) | 140 (54%) | 421 (55%) | 139 (54%) | 138 (51%) | 55 (48%) |
| | DR3/X | 70 (14%) | 23 (9%) | 85 (11%) | 15 (6%) | 19 (7%) | 16 (14%) |
| | DRX/X | 42 (9%) | 45 (18%) | 112 (15%) | 9 (4%) | 10 (4%) | 2 (2%) |
| | Unknown | 0 (0%) | 1 (0%) | 3 (0%) | 0 (0%) | 0 (0%) | 0 (0%) |
| Age of seroconversion | | 5.38 ± .51 | 6.28 ± .64 | 6.93 ± .38 | 3.48 ± .41 | 2.52 ± .25 | 4.71 ± .59 |
| Age of diagnosis | | – | – | – | 7.64 ± .52 | 7.11 ± .54 | 10.46 ± .69 |

Distribution of undiagnosed and diagnosed participants in three trajectories over sex and HLA-DR status; age (mean ± standard deviation) of seroconversion and diagnosis for undiagnosed and diagnosed participants.

The survival curves stratified on HLA-DR status are provided in Supplementary Fig. 4. Survival analysis stratified on HLA-DR status showed no difference in progression between individuals with DR3/4 vs. DR4/X in TR1. In both TR2 and TR3, individuals with DR3/4 progressed faster than those with DR4/X ($p = 0.003$, $p = 0.0041$).

To examine the role of age in progression rates for each trajectory, we separated participants by median age of entry into first IAb positive state (3.75 years). Of note, for TR1-1, there was no difference in survival rates between participants entering the multiple islet autoantibody states before or after the overall median age of entry into first IA positive state ($Z^2(1) = 1.6$, $p = 0.2$) (Fig. 5a). In contrast, participants who entered the first islet autoantibody positive states in TR2 and TR3 did show differences in survival rates by age. Participants entering IAA positive state

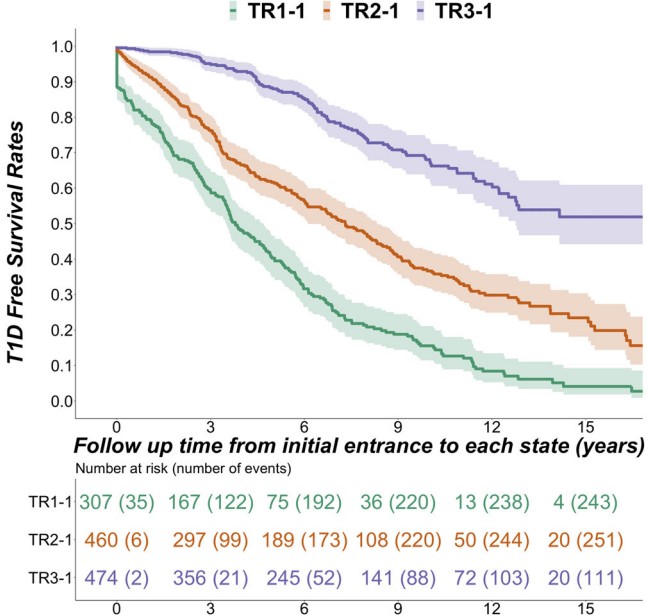

**Fig. 4 Diabetes-free survival stratified by the three trajectories.** T1D-free survival curves (mean and 95% confidence interval) of the three trajectories starting at first entry into IAb positive state (TR1-1, green; TR2-1, red; TR3-1, purple). Source data are provided as a Source Data file.

(TR2-1) earlier than 3.75 years old progressed faster to type 1 diabetes than those entering later ($Z^2(1) = 19.1$, $p < 0.0001$) (Fig. 5b). Similarly, participants entering GADA positive state (TR3-1) before 3.75 years old progressed faster to type 1 diabetes than those entering later ($Z^2(1) = 6$, $p = 0.01$) (Fig. 5c).

Entry into three distinct trajectory states, namely TR1-1, TR2-4, and TR3-2, describe a pattern of high probabilities of GADA and IA-2A positivity, despite similar IAb patterns in each of these states, survival curves showed significant differences in diabetes-free survival rates from those states ($Z^2(2) = 34.1$, $p < 0.0001$). Post-hoc analysis showed significant differences between TR3-2 and TR1-1 ($p < 0.0001$) and between TR3-2 and TR2-4 ($p = 0.007$) (Fig. 5d). TR3-2 showed significantly slower and less frequent progression to type 1 diabetes than TR1-1 and TR2-4. Mean entry ages for GADA and IA-2A positive states in the three trajectories were significantly different ($F(2, 516) = 41.19$, $p < 0.0001$). Tukey HSD test confirmed the mean entry age for TR1-1 as significantly younger than for TR2-4 or for TR3-2. There was no difference in entry age between TR2-4 and TR3-2.

## Discussion

In this study, we discovered 11 latent states of progression to type 1 diabetes onset using data-driven modeling based on longitudinal IAb data. The latent states described three distinct trajectories of disease progression, using an unbiased, probabilistic method and characterized the autoimmune pathways to development of clinical type 1 diabetes. Our descriptions of the dynamic nature of the three trajectories corroborate and expand recent observations from multiple studies[1,3,5–7,13]. The descriptions illustrate the heterogeneous journey of participants defined not just by their first IAb observed but rather by the transitions between probabilistic states of autoantibodies as they age. Our findings suggest age at seroconversion and subsequent progression to type 1 diabetes onset differs significantly among three trajectories. Despite following similar trajectories of IAb patterns, the diagnosed and undiagnosed participants show differences by age, sex, and HLA-DR, at least in the 15 years of follow-up studied. Our longitudinal analysis underscores the necessity of

follow-up beyond cross-sectional description of islet autoantibody positivity as it may not be sufficient to understand an individual's journey towards diagnosis of type 1 diabetes.

The three trajectories found in the present investigation show distinctive progression patterns. The observation that females progress faster than males in TR2 may be related to a more aggressive pathogenesis with age as females tend to be diagnosed with type 1 diabetes earlier than males[14]. The present study discovered underlying subtypes based on three trajectories that could be important in selecting research participants for clinical intervention trials through data-driven modeling. The visual analytic methodology used in this study can be a powerful tool to explore trajectories and to interact with individual-level data, including factors that may vary by location, which could advance clinical research and practice.

The study provides important implications for screening in routine clinical practice, a possibility that is being explored in population screening studies[15,16]. Clinicians may use IAb pattern and age to estimate the trajectory and therefore the risk for developing type 1 diabetes. In other words, our findings show the proportion of participants belonging to a specific trajectory given their age and IAb positivity. Once a likely trajectory is identified, one could examine the preceding and upcoming states for the trajectory, and estimate the type 1 diabetes-free survival of the participants in the trajectory, given their age and IAb positivity.

Another strength of this study is the large number of participants followed from an early age until the appearance of one or more islet autoantibodies. The harmonized data in this international effort not only made it possible to identify and visualize three distinct trajectories but also enabled researchers to examine the impact of different contributing factors specific to the environments of the participants. This approach can also be a valuable addition to available recruitment tools to identify research participants for secondary prevention trials in a variety of settings. Additionally, this study demonstrates the advantages of using interactive visualizations to characterize trajectories and explore data from individuals. By visually representing both the granularity of individual data and the overall patterns of change over time, this method could be combined with other variables to explore new relationships between observational data and identified trajectories. A novel and hitherto uncharted possibility is the ability of visualization not only to delineate groups but also to distinctly follow individuals within trajectories. In clinical applications, this tool may thus have the potential to allow better counseling for individuals and families by providing an improved understanding of likely progression.

Intervention studies have shown differences in response to disease-modifying treatments based on stage of the disease[5,17,18], as well as heterogeneity in response amongst participants[19]. Machine learning models of disease progression combined with interactive visualization tools reveal novel trajectories and enable the requisite increase in granularity needed to support precision medicine approaches to prevention and modulation of disease progression. Future work will include the development of a more directed tool for clinical practice, allowing assessment of an individual patient's progression pattern in the context of population pathways. Future work could also assess the impact of varying genetic backgrounds. By using such information, we can improve our understanding of varying clinical pathways, better utilize resources, and recruit participants following similar disease pathways for clinical interventions.

## Methods

**Participants.** The Type 1 Diabetes Intelligence (T1DI) cohort includes 24,662 participants from five prospective studies (DAISY[20], DiPiS[21], DIPP[22], DEW-IT[23], BABYDIAB[24]) of individuals at increased risk for type 1 diabetes, recruited before

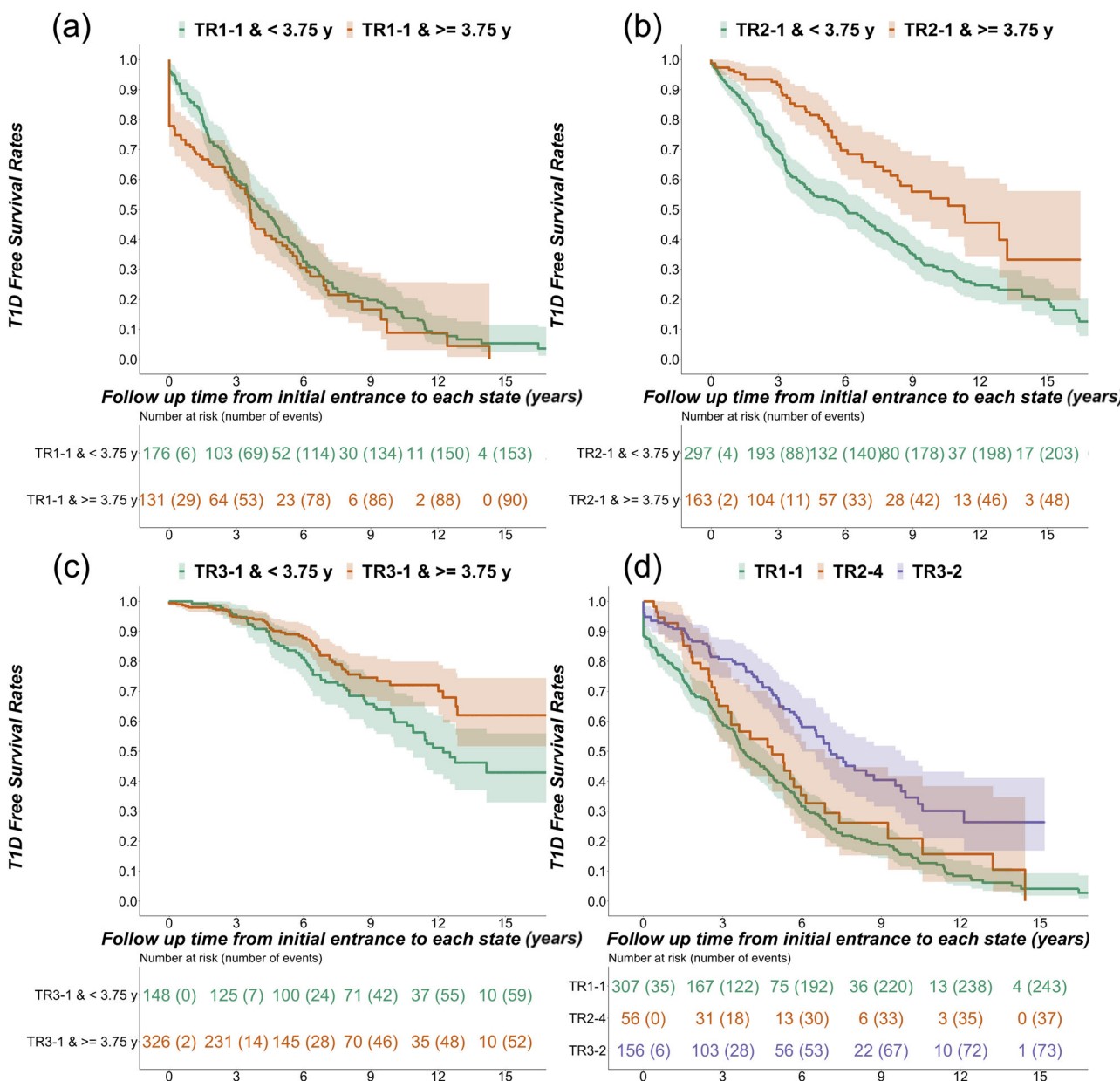

**Fig. 5 Diabetes-free survival stratified by age and trajectory state.** T1D-free survival curves (mean and 95% confidence interval) of participants before or after median age, 3.75 years old, of entry into first IAb positive state (**a** TR1-1, **b** TR2-1, and **c** TR3-1). **d** T1D-free survival curves for individuals in GADA and IA-2 positive states (TR1-1, green; TR2-4, orange; TR3-2, purple). The median ages of entrance for GADA and IA-2A positive states were (i) TR1-1: 3.0; (ii) TR2-4: 7.6; (iii) TR3-2: 7.1 years old. Source data are provided as a Source Data file.

age 25 years based on high-risk HLA genotype or history of first-degree relative with type 1 diabetes. Presence or absence of three islet autoantibodies, GADA, IAA, and IA-2A, were combined across studies, and data were included up to a follow up of 15 years or until the diagnosis of type 1 diabetes, whichever came first, per originating study protocols. Type 1 diabetes was diagnosed according to American Diabetes Association criteria[25], seroconversion by two consecutive visits with at least one IAb persistently positive at both visits and seroconversion age as the first of the two visits. In addition to IAb measurements and outcome of diagnosis, the T1DI cohort contains anthropometric, metabolic, diet, and environmental measurements. All uses of human materials from the individual studies - BABYDIAB, DAISY, DEW-IT, DiPiS, and DIPP were each approved by their relevant ethics review boards: (i) BABYDIAB by the Bavarian ethical committee (Bayerische Landesärztekammer; No. 95357); (ii) DAISY by the Colorado Multiple Institutional Review Board (COMIRB; COMIRB 92-080); (iii) DEW-IT by WIRB-Copernicus Group Institutional Review Board (WCG-IRB; DSHS Project B-092199-H); (iv) DiPiS by the regional ethics board in Lund Sweden (Dnr LU-490-99); Joint Municipal Authority of Northern Ostrobothnia Hospital District, Regional Ethics Committee, Finland. All data and samples were collected under the written informed consent of the participants or their legal parents or guardians.

**Modeling analysis**. Using a probabilistic approach[26], incorporating Continuous-Time Hidden Markov Model (CT-HMM)[27], we trained disease progression models (DPM) on presence or absence of IAb and the age of participants. The DPMs discovered latent states from longitudinal measurements of the three IAbs from each participant's visit and the age of the participant at the visit (Supplementary Table 3 and Supplementary Fig. 6 for examples of the observational data)[28]. Further analyses correlated latent states and resulting trajectories with other study variables.

The DPM were generated in an unsupervised way, meaning type 1 diabetes diagnoses were not used to inform model parameters[29]. To produce robust results, model parameters (for CT-HMM) were learned in 1900 repeated experiments in total: 100 sub-samples (bootstrapping) × 19 different latent state models, exploring possible numbers of latent states ranging from 2 to 20-state models. For each latent state model experiment, we randomly split data in the ratio of 70:30 (training: held-out validation test) for model training and model validation[29]. In the random split, each participant's entire visit history could either belong to the training set or the test set, but not both, thus creating random sub-samples of the observed data. In each experiment, the model parameters for each of the 19 possible models with a number of latent states ranging from 2 to 20 were discovered from the time-

stamped (age at visit) observational data (training set) consisting of the IAbs. The CT-HMM parameters (transition and emission probabilities) were learned by maximum likelihood estimation using the Expectation-Maximization (EM) algorithm iteratively till convergence and the number of iterations were empirically determined based on computed likelihood in each iteration. At convergence, the trained model assigned a latent state number to each participant visit (indexed by age) in the training set. Similarly, the participant visits in the held-out test set were assigned a model (latent) state number or "labeled" in these experiments[26]. We also calculated the (predictive) log-likelihood of observed data given the model using the held-out set. The predictive log-likelihood was used to select the best model we use for the analysis in the manuscript.

To learn a robust model for disease progression, we only included participants eventually diagnosed with type 1 diabetes (within 15 years of follow-up) and who had three or more visits during the follow-up period. Additionally, the model was learned using data from only three T1DI studies (DAISY, DIPP, DiPiS) ($n = 559$) for whom data were available at the time of model development. Later, independent model validation was done using participants from two other studies (BABYDIAB, DEW-IT) ($n = 150$). Since, we performed 1900 experiments, each generating a possible disease progression model (of 2 to 20 latent states), we needed to select a model based on the best fit among the latent states. To find the best model fit among the latent states explored, we computed the Bayesian Information Criterion (BIC) score[30]. BIC penalizes model overfit (i.e. number of model parameters to learn given the number of latent states and the number of observations required for training). We selected the most probable model from a set of competing models having minimal BIC scores (latent state model 11, 12, 13) and the highest value for predictive log-likelihood (calculated based on a held-out test set during the learning process).

The final model contains 11 latent states representing the observed islet autoimmunity development of diagnosed participants from the T1DI cohort. It was used to draw insights from all participants with IAb positivity in the analysis cohort, i.e., irrespective of their diagnosis. Longitudinal observations of participants were labeled using the model for further analysis. Specifically, the 11-state model was used to label each participant's visit with one of the 11 states using an index ranging from 0 to 10. The results show that most participants started and ended their observations within one out of three trajectories: Trajectory 1 consists of three states (0,1,2), starting from the state "0"; Trajectory 2 consists of five states (3,4,5,6,7), starting from state "3"; Trajectory 3 consists of three states (8,9,10), starting from the state "8", as Supplementary Fig. 7 illustrates. The analysis of state characteristics revealed that the starting states 0 (TR1-0), 3 (TR2-0), and 8 (TR3-0) were characterized by low probabilities of antibody positivity (Figs. 1 and 2 in the manuscript). As the manuscript describes, each trajectory is characterized by the first state with autoantibody positivity, such as multiple islet autoantibodies (Trajectory 1), IAA (Trajectory 2), GADA (Trajectory 3). To clearly describe the distinct patterns of the three trajectories in the manuscript, we renamed the 11 states to the {Trajectory Name—Step within Each Trajectory} format, e.g., TR1-1, in the manuscript. In this way, readers can recognize which trajectory and which step a participant's visit belongs to by the name. These participants' data were then divided into those who developed type 1 diabetes (Diagnosed/D) during the study period vs. those who did not, or were lost to follow-up, (Undiagnosed/UD).

**Analysis cohort**. We studied 2172 individuals from the T1DI cohort with one or more IAb measurements at or before the age of 2.5 years and at least one positive IAb measurement during participation, identified as "diagnosed" ($n = 652$), or "undiagnosed" ($n = 1520$), based on the diagnosis status at their last observation. The median age at participants' last observation were 7.62 and 12.87 years for the diagnosed and undiagnosed, respectively (see Supplementary Fig. 2 in Supplementary Information for further detail). On visualization, T1DI-DPM discovered three trajectories, which uniquely fit all but 27 participants (1.2%, nine diagnosed, 18 undiagnosed), who could possibly fit into two different trajectories. After eliminating these 27 individuals, our analytic cohort included 2145 participants (98.8%, 643 diagnosed, 1502 undiagnosed). A flow chart of the cohort selection process and criteria is in Supplementary Fig. 1 in Supplementary Information.

The data that support the findings of this study are available on request from the corresponding author B.K. The data are not publicly available due to privacy concerns.

**Analysis methods**. We used an interactive visualization method called DPVis[29] to discover and characterize trajectories in the IAb positive participants by enabling visual identification and analysis of patterns of IAb trajectories. Using the visually discovered trajectories as boundaries for groups, we performed a one-way, two-sided ANOVA followed by Tukey HSD for statistical differences in age at seroconversion and age at diabetes onset. Two-sided Chi-Square test was used to examine the relationship between trajectories and participant characteristics, specifically HLA-DR status and sex. We performed Kaplan–Meier survival analysis and tested differences in type 1 diabetes-free survival rates between trajectories using the two-sided log-rank test. We then compared the survival rates within each trajectory before or after median age of entry into first IAb state, and finally compared the survival rates after entering GADA and IA-2A positive states in each trajectory.

**Reporting summary**. Further information on research design is available in the Nature Research Reporting Summary linked to this article.

## Data availability

The raw data in this study have been separated and deposited in each of the five study groups: DiPiS, BABYDIAB, DIPP, DEW-IT, and DAISY. The raw data are protected and are not publicly available due to data privacy laws. The source data for figures generated in this study are provided in the Source Data file. All other data that support the findings of this study are included in Supplementary Information or can be made available upon reasonable request. Source data are provided with this paper.

## Code availability

The code to generate the waterfall diagram is deposited in the following repository (https://github.com/bckwon/dpvis-waterfall). All other figures can be generated using any standard charting library.

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

## Acknowledgements

We wish to thank the T1DI Study Group for their help in this work. The T1DI Study Group consists of following members: (1) JDRF—J.L.D., Olivia Lou, and Frank Martin; (2) IBM—V.A., Mohamed Ghalwash, E.K., B.C.K., Ying Li, Zhiguo Li, Bin Liu, Ashwani Malhotra, Shelley Moore, and K.N.; (3) DiPiS—Helena Elding Larsson, Josefine Jönsson, Å.L., M.L., Marlena Maziarz, and Lampros Spiliopoulos; (4) BABYDIAB—P.A., Christiane Winkler, and Anette Ziegler; (5) DIPP—Heikki Hyöty, Jorma Ilonen, Mikael Knip, J.T., and R.V.; (6) DEW-IT—Bill Hagopian, Michael Killian, and Darius Schneider; (7) DAISY—B.I.F., Jill Norris, Marian Rewers, Andrea Steck, Kathleen Waugh, and Liping Yu. We wish to thank the members of the T1DI Study Group Scientific Advisory Board (Richard Oram, University of Exeter, UK, Eoin Mckinney, University of Cambridge, UK, Bobbie-Jo Webb-Robertson, PNNL, USA, Nitesh Chawla, University of Notre Dame, USA, Soren Brunak, University of Copenhagen, DE, Len Harrison, WEHI, AU) for their invaluable input over the course of this study. This work was supported by funding from JDRF (IBM: 1-RSC-2017-368-I-X, 1-IND-2019-717-I-X), (DAISY: 1-SRA-2019-722-I-X, 1-RSC-2017-517-I-X, 5-ECR-2017-388-A-N), (DiPiS: 1-SRA-2019-720-I-X, 1-RSC-2017-526-I-X), (DIPP: 1-RSC-2018-555-I-X), (DEW-IT: 1-SRA-2019-719-I-X, 1-RSC-2017-516-I-X) as well as NIH (DAISY: DK032493, DK032083, DK104351, and DK116073); DiPiS: DK26190 and the CDC (DEW-IT: UR6/CCU017247). The DIPP study was funded by JDRF (grants 1-SRA-2016-342-M-R, 1-SRA-2019-732-M-B); European Union (grant BMH4-CT98-3314); Novo Nordisk Foundation; Academy of Finland (Decision No 292538 and Centre of Excellence in Molecular Systems Immunology and Physiology Research 2012-2017, Decision No. 250114); Special Research Funds for University Hospitals in Finland; Diabetes Research Foundation, Finland; and Sigrid Juselius Foundation, Finland. The BABYDIAB study was funded by the German Federal Ministry of Education and Research to the German Center for Diabetes Research. The DiPiS study was funded by Swedish Research Council (grant no. 14064), Swedish Childhood Diabetes Foundation, Swedish Diabetes Association, Nordisk Insulin Fund, SUS funds, Lion Club International, district 101-S, The royal Physiographic society, Skåne County Council Foundation for Research and Development as well as LUDC-IRC/EXODIAB funding from the Swedish Foundation for Strategic Research (Dnr IRC15-0067) and Swedish research council (Dnr 2009-1039). Additional funding for DEW-IT was provided by the Hussman Foundation and by the Washington State Life Science Discovery Fund.

## Author contributions

B.C.K., V.A., P.A., J.L.D., W.H., J.H., E.K., A.L., M.L., K.N., J.T., R.V., and B.I.F. wrote the manuscript. All authors, who are members of the T1DI Study Group, contributed to discussions and critical editing of the manuscript.

## Competing interests

B.C.K., V.A., J.H., E.K., K.N. are currently employees of IBM Research. J.L.D. is currently an employee of Janssen Research & Development, LLC, however, the work was done while she was at JDRF. The remaining authors declare no competing interests.

## Additional information

## the T1DI Study Group

Bum Chul Kwon [1,10 ✉], Vibha Anand [1,10], Peter Achenbach [2], Jessica L. Dunne [3], William Hagopian [4], Jianying Hu [5], Eileen Koski [5], Åke Lernmark [6], Markus Lundgren [6], Kenney Ng [1], Jorma Toppari [7], Riitta Veijola [8] & Brigitte I. Frohnert [9]

