## [Peer Review File · Nature Communications]

Progression of type 1 diabetes from latency to symptomatic disease is predicted by distinct autoimmune trajectoriesREVIEWER COMMENTS

Reviewer #1 (Remarks to the Author):

Review of:

Visualizing distinct islet autoimmune trajectories towards type 1 diabetes

In this article the authors use longitudinal data on presence or absence of autoantibodies and Hidden Markov Models (HMM) to discover latent states and applied visualizations to understand islet 34 autoimmunity trajectories from these latent states. Then, they evaluated the evolution of autoantibody trajectories combined with participants' clinical characteristics both 36 statistically and visually.

The article is very interesting and uses state-of-the-art strategies to try to tackle the problem of determining T1D progression. Nevertheless...

Observations:

1) It is necessary to allow access to the input data used to generate the trajectories. At least a blind version in which only mask individual ID's are reported. It is difficult to understand the input data since it is not clearly explained in the article.

By the following statement:

"Presence or absence of three islet autoantibodies, GADA, IAA and IA-2A, were combined across studies, and data were included up to a follow up of 15 years or until diagnosis of type 1 diabetes, whichever came first, per originating study protocols."

I assume the data is a table in which in rows there are several sample ID's per each individual and three columns of binary (presence/absence) of the three (GADA, IAA, IA2A) autoantibodies evaluated, is this true?

2) If the previous assumption is true it is necessary to indicate how the samples were synchronized. Is it in terms of age of the individual at the moment they took the sample? This has to be clearly explained in the methodology section.

3) The article indicates that 1900 experiments were performed per each range of pre-established latent states ranging from 2 to 20.

It is necessary to specify if these latent states models were based on different visits/blood samples.

4) If this is true it is again important to indicate how these blood samples were synchronized in terms of the individuals. For instance, the second visit of individual 1 could occur at 2 years of age while the second visit of individual 2 could occur at 6 months of age.

5) Perhaps the individuals were synchronized in terms of age in months or age in years? If this is the case what is the number (N) of individuals in each particular latent state in order for you to determine that it is significantly enough to be considered as a real latent state.

6) What is the relationship among the 11 latent states and the 2 steps in trajectory 1, 4 steps in trajectory 2 and 2 steps in trajectory 3. In other words what is the rationale behind mentioning 11 latent states but then showing only 2, for instance, in Trajectory 1.

7) These steps in each trajectory are not well described in the article. In other words, it is necessary to indicate what is occurring for you to determine that a transition between TR1-0 and TR1-1 has occurred.

8) How many latent states are included in each trajectory step? And why?

9) The methodology in this sense, has to be more explicit.

For instance, in the following statement:

“For each latent state model, we randomly split data 70:30 (training: held-out test) for model validation²⁷. In each experiment, the model parameters for the given set of latent states were discovered from time-stamped observational IAb sequences (training set). Then the observed IAb sequences in the held-out test set were assigned a model (latent) state number or “labeled.””

What exactly are the “observed IAb sequences”? What do sequences mean in this context?

This has to be clearly explained in the methodology section.

Reviewer #2 (Remarks to the Author):

This study reports three novel islet autoimmunity trajectories preceding progression to type 1 diabetes in children by using longitudinal data from 2145 islet autoantibody positive participants and machine learning methods. Relationship between the three different trajectories was further defined by age (seroconversion first time and at diagnosis), sex, HLA-DR status, and progression to type 1 diabetes.

These novel results are of interest and potentially clinically valuable. I have only some minor comments

Main manuscript:

- line 144, missing a p-value for comparison between TR1 and TR2
- line 151, missing a p-value. Could be added as a footnote to the table if preferred.
- line 157, probably a typo, should it be TR2?
- line 164-167, I would prefer that you report the significant findings rather than non-significant.
- line 169, I believe this is the tables S1 (S4-1) and S2 (S4-2), please correct. And please be consistent on how you refer to the supplementary tables and figures. Use either the section-heading or the number of the table and figure.
- line 182-183, missing p-value.
- line 187, reported data are different from supplement, please check.
- line 191, reported p-value are not reported in the supplement, please add these data in the supplement as well.
- line 193, please check the p-value against supplement.

Supplement:

- Figure S3 and Table S3 in Section S5-1 is not referred to in the main manuscript

Additional comments regarding survival curves in main manuscript and supplement:

- Please use larger font in the tables number at risk, difficult to read. Enhance the line in the survival curve, difficult to see amongst all the colors.

Responses to the Reviewers' Comments

Please note that in this response letter we refer to the line numbers as shown in the revised version so that reviewers can find the revision easily. We highlighted the Reviewers' questions/requests/statements in red.

Reviewer #1 (Remarks to the Author):

1) It is necessary to allow access to the input data used to generate the trajectories. At least a blind version in which only mask individual ID's are reported. It is difficult to understand the input data since it is not clearly explained in the article.

>>> We appreciate the Reviewer's concern and request for more detail to better understand the structure and content of the data analyzed in this manuscript. For the reviewers and readers to better understand the data structure, we prepared the table below that illustrates representative input data as well as the corresponding output column from HMM analysis (in bold). To generate the trajectories, we used four input data variables: GADA, IAA, IA-2A (presence of islet autoantibody at visit =1; absence = 0), and Age at visit for a series of samples (visits) per participant. Next, the Hidden Markov Model is trained with the input data. As a result of the training, each visit is labelled with a latent state, as illustrated in the HMM State column (in bold). The final model has 11 states, so the model assigns a number between 0 and 10 for each visit. These 11 states were later renamed using the three identified trajectories, denoted TRx-y (in parentheses), where x indicates trajectory 1, 2 or 3 and y indicates the states for that trajectory. We describe the renaming process in more detail in response to query (6) below.

We have added the following table (Table S3) to the supplement and referred to it in the method section (page 13, line 324-326).

Participant ID	Model Input				Model Output
	GADA	IAA	IA-2A	Age (Y)	HMM State
Participant 1	0	0	0	0.5	3 (TR2-0)
Participant 1	0	1	0	1.0	4 (TR2-1)
Participant 1	0	1	0	1.6	4 (TR2-1)
Participant 1	0	1	1	3.3	5 (TR2-2)
Participant 1	1	1	1	4.7	6 (TR2-3)
Participant 1	1	1	1	6.2	6 (TR2-3)
Participant 1	1	0	1	7.0	7 (TR2-4)
.....					
Participant 2	0	0	0	0.75	0 (TR1-0)
Participant 2	0	0	0	1.55	0 (TR1-0)
Participant 2	0	0	0	1.8	0 (TR1-0)
Participant 2	1	1	1	2.7	1 (TR1-1)
Participant 2	1	1	1	3.5	1 (TR1-1)
Participant 2	1	0	1	4.1	1 (TR1-1)
Participant 2	1	1	1	5.0	1 (TR1-1)
Participant 2	1	1	1	6.1	1 (TR1-1)

Participant 2	1	1	1	7.0	1 (TR1-1)
Participant 2	0	0	1	7.9	2 (TR1-2)
Participant 2	0	0	1	8.8	2 (TR1-2)
.....					
Participant 3	0	0	0	1.9	8 (TR3-0)
Participant 3	0	0	0	3.6	8 (TR3-0)
Participant 3	1	0	0	6.3	9 (TR3-1)
Participant 3	1	0	0	7.6	9 (TR3-1)
Participant 3	1	0	1	8.1	10 (TR3-2)

We also offer access to source data from the US sites that are not covered by the European privacy regulations (GDPR, national regulations/requirements). The data file (nature_communications_data.csv) is being submitted to the system along with other files. This will give the Reviewer wider insight into the data structure and allow an opportunity to evaluate the disease modelling. We believe this dataset is likely to generate similar results in terms of overall grouping of the participants into three trajectories as well as the observations regarding these groups. We would caution the Reviewer that this exercise would of course not yield the exact results that we report in the paper which were generated from the full combined source data. Should the reviewer feel that access to the full dataset is critical to evaluation, we can provide this after the reviewer has signed a confidentiality agreement. This would, per necessity require unblinding of the reviewer to the study team.

By the following statement:

“Presence or absence of three islet autoantibodies, GADA, IAA and IA-2A, were combined across studies, and data were included up to a follow up of 15 years or until diagnosis of type 1 diabetes, whichever came first, per originating study protocols.”

I assume the data is a table in which in rows there are several sample ID's per each individual and three columns of binary (presence/absence) of the three (GADA, IAA, IA2A) autoantibodies evaluated, is this true?

>>> Yes, the Reviewer is correct. In addition, there is the age of the individual in years at the time of the visit as shown in the example table above (Table S3).

2) If the previous assumption is true it is necessary to indicate how the samples were synchronized. Is it in terms of age of the individual at the moment they took the sample? This has to be clearly explained in the methodology section.

>>> We thank the Reviewer for pointing this out. Here we clarify the modeling approach we used in the manuscript. We used a Continuous-Time Hidden Markov Model (CT-HMM) (Lange and Minin, 2013) to model the disease progression. The CT-HMM takes a continuous time variable as input and can directly handle irregularly sampled temporal data. As a result, no time synchronization of the samples was needed. In the manuscript, we revised the method section to clarify the specific HMM model we used (page 2, line 32-33; page 4, line 76; page 13, line 321-322; page 13, line 330; page 13, line 339). We have added the following reference to the revised manuscript:

Jane M. Lange and Vladimir N. Minin. Fitting and interpreting continuous-time latent Markov models for panel data. *Statistics in Medicine*, 32(26):4581–4595, November 2013.

3) The article indicates that 1900 experiments were performed per each range of pre-established latent states ranging from 2 to 20.

It is necessary to specify if these latent states models were based on different visits/blood samples.

>>> The dataset we used for training the models (training set) was a subset of the dataset we used for analysis (analysis cohort). We used the data of 559 participants from three T1DI study groups (DAISY, DIPP, DiPiS) for model development and used a different 150 participants from two other studies (BABYDIAB, DEW-IT) for validation.

The samples for the model development set were randomized into 10 “experimental” sets where each set was randomly split on subject-visits (in a 70-30 ratio) into “training” and “held-out” sets. For each of the 10 experimental sets, we performed 10 runs (with different random initializations) to learn 19 different models (with the number of states K , ranging from 2 to 20). These 1900 experiments (10x10x19) generated 1900 model instances. We selected the model with the best predictive Log-likelihood and Bayesian information criterion (BIC) score as the disease progression model (DPM) to use for downstream analysis.

We then used the selected DPM model (i.e., the 11-state model), to label the visits of the entire analysis cohort (N=2145).

We have substantially revised the method section (page 13 – 14, line 328 – 356) to clarify the model construction and validation processes.

4) If this is true it is again important to indicate how these blood samples were synchronized in terms of the individuals. For instance, the second visit of individual 1 could occur at 2 years of age while the second visit of individual 2 could occur at 6 months of age.

>>> As mentioned above in the response to comment 2, we did not need to synchronize the samples of participants because our modeling approach used CT-HMM.

5) Perhaps the individuals were synchronized in terms of age in months or age in years? If this is the case what is the number (N) of individuals in each particular latent state in order for you to determine that it is significantly enough to be considered as a real latent state.

>>> As mentioned above in the response to comment 2, we did not need to synchronize the samples of participants because we used CT-HMM. Figures 1 & 2 show the number of unique participants (the “N” column) associated with each latent state. The number of visits and the number of unique participants for each state is summarized in the table below:

States	Number of visits	Number of unique participants
--------	------------------	-------------------------------

		(Diagnosed and Undiagnosed)
TR1-0	8528	690
TR1-1	2168	307
TR1-2	1395	114
TR2-0	4238	419
TR2-1	3144	460
TR2-2	1063	226
TR2-3	1531	134
TR2-4	823	56
TR3-0	12233	818
TR3-1	4446	474
TR3-2	2640	156

6) What is the relationship among the 11 latent states and the 2 steps in trajectory 1, 4 steps in trajectory 2 and 2 steps in trajectory 3. In other words what is the rationale behind mentioning 11 latent states but then showing only 2, for instance, in Trajectory 1.

>>> Based on our modeling results from the data, each participant may only go through a subset of the 11 states because of the modeling constraints imposed. For example, a state transition can only occur from one state (n) to the next (n+1) state in the forward direction. However, a participant's first visit could be labelled to any of the available states, but then could only transition one step forward subsequently. To capture the heterogeneous nature of antibody progression, we did not impose any constraint on how many states each patient must traverse, rather we left it to the machine learning model to determine, based on the observational data, both the states and the assignment of each visit to a state. As a result, our model naturally discovered three trajectories, each consisting of states 0-2, 3-7, 8-10 respectively, as shown in the figure below (Figure S7). Using this approach, 98.8% of participants fit into one of the three discovered trajectories, meaning they start and end their state transitions within each trajectory, as Section S1 describes. As written in the manuscript, we removed the remaining 1.2% of participants who did not fit into one of the three trajectories from the analysis cohort.

Here we provide a reason why we renamed the name of the 11 states from a range between 0 and 10 to the TRX-Y format. For the analysis, a single 11-state CT-HMM model was trained and then used to label each participant's visit with one of the 11 states (using an index ranging from 0 to 10 as the "HMM state" column in the table in to comment 1) above and the Figure above shows. In modeling the disease progression, we allow a participant's first visit to be mapped to any of the states and then progress from there. As the figure above illustrates, in the resulting model, we observed that most participants started and ended their encounters within one of three trajectories that emerged from the training process: Trajectory 1 consists of three states (0,1,2), starting from state 0; Trajectory 2 consists of five states (3,4,5,6,7), starting from state 3; Trajectory 3 consists of three states (8,9,10), starting from state 8. The analysis of state characteristics revealed that the starting states 0 (TR1-0), 3 (TR2-0) and 8 (TR3-0) were characterized by low probabilities of antibody positivity (see Figure 1 & 2 in the manuscript). As the manuscript describes, each trajectory is characterized by the first state with autoantibody positivity, such as multiple islet autoantibody (Trajectory 1), IAA (Trajectory 2), GADA (Trajectory 3). To describe the distinct patterns of the three trajectories more clearly in the manuscript, we renamed the 11 states to the {Trajectory Name – Step within Each Trajectory} format, such as TR1-0, TR2-3, TR3-1. In this way, readers can recognize which trajectory and which step a participant's visit belongs to by the name.

We revised L366 - 390 in manuscript. We added the new figure (Figure S7) to the supplement.

7) These steps in each trajectory are not well described in the article. In other words, it is necessary to indicate what is occurring for you to determine that a transition between TR1-0 and TR1-1 has occurred.

>>> The trajectory steps correspond to state transitions in the CT-HMM model. The state sequences are determined in a data-driven manner using the Viterbi algorithm. The objective function of the algorithm is to maximize the overall fitness (not local fitness) of the model to the entire sequence of observations per participant. When the transition from one state to another occurs is determined by the combination of change in the underlying distribution of input variables, in this case the constellation of positive antibodies in the context of age at visit, and the transition probability between the two states learned from data. The distinct characteristics of distributions of islet autoantibodies show the manifestation of each state. For instance, TR2-0 to TR2-1 is likely to occur when a participant acquires IAA. The overall pattern of antibody positivity is how we report the interpretation of each state, e.g., TR1-0 as IAb negative, TR1-1 as multiple IAb positive, TR1-2 as IA-2A positive.

8) How many latent states are included in each trajectory step? And why?

>>> As explained in the response to comment 6 above, one trajectory step maps to one latent state. The number of latent states, the probability distributions that described each state, and the transition probabilities between states are all determined by the training process based on data.

9) The methodology in this sense, has to be more explicit. For instance, in the following statement:

“For each latent state model, we randomly split data 70:30 (training: held-out test) for model validation²⁷. In each experiment, the model parameters for the given set of latent states were discovered from time-stamped observational IAb sequences (training set). Then the observed IAb sequences in the held-out test set were assigned a model (latent) state number or “labeled.””
 What exactly are the “observed IAb sequences”? What do sequences mean in this context?
 This has to be clearly explained in the methodology section.

>>> We appreciate the reviewer’s request for further clarity and detail to describe the methods in this manuscript and have revised the Methods section to provide more clarity.

The time-stamped, observed IAb sequences refer to **the input data table** which includes three islet autoantibodies and age at visit collected per individual participant. The input table is illustrated in the response to comment 1 above (Table S3). We also provide a graphical illustration of the data structure in Figure S6 below. It shows 3 participants that have varying numbers of visits with irregular time intervals. Each visit includes a tuple of measurements: three islet autoantibody positivity indicators (GADA, IAA, and IA-2A) and age at visit. The observed IAb sequence of the individual in the held-out dataset was “labeled” with the appropriate series of states based on the temporal sequence of the antibody pattern and age at visit. The model validation process is described in [28].

We have added the following illustration to the supplement as well (Figure S6).

Reviewer #2 (Remarks to the Author):

Main manuscript:

-line 144, missing a p-value for comparison between TR1 and TR2

>>> We added the missing p-value in the manuscript (Lines 149-151).

-line 151, missing a p-value. Could be added as a footnote to the table if preferred.

>>> We added the missing p-value in the manuscript (Lines 157-159).

-line 157, probably a typo, should it be TR2?

>>> We thank the reviewers for their careful reading of the manuscript. The reviewer is correct, this should be TR2. We corrected it accordingly (Line 165).

-line 164-167, I would prefer that you report the significant findings rather than non-significant.

>>> We revised the paragraph so that we report the significant findings and omit non-significant findings (Lines 171-175).

-line 169, I believe this is the tables S1 (S4-1) and S2 (S4-2), please correct. And please be consistent on how you refer to the supplementary tables and figures. Use either the section-heading or the number of the table and figure.

>>> Yes, it should be Table S1 and S2. We thank the reviewer for this suggestion and have made changes so that we refer to tables and figures directly (Line 186).

-line 182-183, missing p-value.

>>> We added the missing p-value in the manuscript (Lines 200-201).

-line 187, reported data are different from supplement, please check.

>>> We thank the reviewer for noting this discrepancy. The reported data in the main text is correct. We modified the text in the supplement.

-line 191, reported p-value are not reported in the supplement, please add these data in the supplement as well.

>>> We verified and revised the text so that the values are consistent in the manuscript and the supplement.

-line 193, please check the p-value against supplement.

>>> The reviewer is correct, we checked the p-value and corrected it (Line 217).

Supplement:

-Figure S3 and Table S3 in Section S5-1 is not referred to in the main manuscript

>>> We removed Figure S3 and Table S3 because they are not discussed in the main manuscript.

Additional comments regarding survival curves in main manuscript and supplement:

-Please use larger font in the tables number at risk, difficult to read. Enhance the line in the survival curve, difficult to see amongst all the colors.

>>> We appreciate this feedback and have enlarged the font size in the risk table, enhanced the thickness of the lines in survival curves, and chose a lighter tone for the confidence intervals to maintain the high contrast.

REVIEWERS' COMMENTS

Reviewer #1 (Remarks to the Author):

After a very thorough review, I have no further questions or concerns. This work is very interesting and I approve this version for publication.

Reviewer #2 (Remarks to the Author):

I thanks the authors of this publication for taking our considerations in to account. I have no further comments.